# Disruption of stromal hedgehog signaling initiates RNF5-mediated proteasomal degradation of PTEN and accelerates pancreatic tumor growth

Jason R Pitarresi[2,16,*], Xin Liu[2,17,*], Alex Avendano[3], Katie A Thies[1], Gina M Sizemore[4], Anisha M Hammer[2], Blake E Hildreth III[1] (ID), David J Wang[5], Sarah A Steck[4], Sydney Donohue[6], Maria C Cuitiño[1,2], Raleigh D Kladney[6], Thomas A Mace[7], Jonathan J Chang[3], Christina S Ennis[3], Huiqing Li[8], Roger H Reeves[8] (ID), Seth Blackshaw[9], Jianying Zhang[10], Lianbo Yu[10], Soledad A Fernandez[10], Wendy L Frankel[11], Mark Bloomston[12], Thomas J Rosol[13], Gregory B Lesinski[14], Stephen F Konieczny[15], Denis C Guttridge[2,5], Anil K Rustgi[16], Gustavo Leone[1,2], Jonathan W Song[3] (ID), Jinghai Wu[6], Michael C Ostrowski[1,2] (ID)

The contribution of the tumor microenvironment to pancreatic ductal adenocarcinoma (PDAC) development is currently unclear. We therefore examined the consequences of disrupting paracrine Hedgehog (HH) signaling in PDAC stroma. Herein, we show that ablation of the key HH signaling gene *Smoothened* (*Smo*) in stromal fibroblasts led to increased proliferation of pancreatic tumor cells. Furthermore, *Smo* deletion resulted in proteasomal degradation of the tumor suppressor PTEN and activation of oncogenic protein kinase B (AKT) in fibroblasts. An unbiased proteomic screen identified RNF5 as a novel E3 ubiquitin ligase responsible for degradation of phosphatase and tensin homolog (PTEN) in *Smo*-null fibroblasts. *Ring Finger Protein 5* (*Rnf5*) knockdown or pharmacological inhibition of glycogen synthase kinase 3β (GSKβ), the kinase that marks PTEN for ubiquitination, rescued PTEN levels and reversed the oncogenic phenotype, identifying a new node of PTEN regulation. In PDAC patients, low stromal PTEN correlated with reduced overall survival. Mechanistically, PTEN loss decreased hydraulic permeability of the extracellular matrix, which was reversed by hyaluronidase treatment. These results define non-cell autonomous tumor-promoting mechanisms activated by disruption of the HH/PTEN axis and identifies new targets for restoring stromal tumor-suppressive functions.

## Introduction

The most prominent histopathological hallmark of pancreatic cancer is its uniquely dense tumor stroma, comprised activated fibroblasts, immune cell infiltrates, abnormal angiogenesis, and extracellular matrix (ECM) (Feig et al, 2012). The stroma undergoes a dramatic expansion in concert with the step-wise development of pancreatic ductal carcinoma (PDAC), suggesting that the stroma is an active partner in PDAC initiation and progression (Feig et al, 2012). In support of this view, a cohort of patients with tumors exhibiting a higher content of smooth-muscle actin (SMA)–positive cells had significantly reduced overall survival compared with individuals with fewer of these cells (Fujita et al, 2010). However, recent results have challenged this interpretation, demonstrating in mouse models that ablation of fibroblasts in the pancreatic tumor stroma increases tumor growth and, contrary to previous studies, an independent cohort of patients with tumors having fewer SMA-positive cells had poorer overall survival than those more enriched for SMA-positive cells (Ozdemir et al, 2014).

Molecular and genomic analysis of human pancreatic tumors identified Hedgehog (HH) signaling as a core pathway contributing to tumor malignancy (Jones et al, 2008; Tian et al, 2009). A prevailing hypothesis is that HH signaling in pancreatic cancer occurs in

[1]Hollings Cancer Center and Department of Biochemistry & Molecular Biology, Medical University of South Carolina, Charleston, SC, USA   [2]Ohio State Biochemistry Graduate Program, The Ohio State University Columbus, Columbus, OH, USA   [3]Department of Mechanical and Aerospace Engineering and Ohio State Comprehensive Cancer Center, The Ohio State University, Columbus, OH, USA   [4]Department of Radiation Oncology and Ohio State Comprehensive Cancer Center, The Ohio State University, Columbus, OH, USA   [5]Hollings Cancer Center and the Darby Children's Research Institute, Medical University of South Carolina, Charleston, SC, USA   [6]Cancer Biology & Genetics Department and Ohio State Comprehensive Cancer Center, The Ohio State University, Columbus, OH, USA   [7]Department of Internal Medicine, The Ohio State University, Columbus, OH, USA   [8]Department of Physiology and McKusick-Nathans Institute for Genetic Medicine, Johns Hopkins University School of Medicine, Baltimore, MD, USA   [9]Solomon H. Snyder Department of Neuroscience, Johns Hopkins University School of Medicine, Baltimore, MD, USA   [10]Department of Biomedical Informatics' and Center for Biostatistics, The Ohio State University, Columbus, OH, USA   [11]Department of Pathology, The Ohio State University, Columbus, OH, USA   [12]Department of Surgery, The Ohio State University, Columbus, OH, USA   [13]Department of Biomedical Sciences, Ohio University, Athens, OH, USA   [14]Department of Hematology & Medical Oncology and Winship Cancer Institute, Emory University, Atlanta, GA, USA   [15]Department of Biological Sciences, Purdue Center for Cancer Research and Bindley Bioscience Center, Purdue University, West Lafayette, IN, USA   [16]Division of Gastroenterology, Department of Medicine and Abramson Cancer Center, University of Pennsylvania, Philadelphia, PA, USA   [17]Department of Surgery, Stanford University, Stanford, CA, USA

Correspondence: ostrowsk@musc.com
*Jason R Pitarresi and Xin Liu contributed equally to this work.

 

a paracrine manner leading primarily to activation of the pathway in stromal fibroblasts (Bailey et al, 2009; Tian et al, 2009). Tumor-derived HH ligands, such as sonic hedgehog (SHH), bind to their cognate receptor Patched1 (PTCH1) on stromal fibroblasts, releasing its repression of Smoothened (SMO) and allowing for activation of downstream glioma-associated oncogene homolog (GLI) transcription factors. Pre-clinical studies suggested that inhibition of canonical stromal HH signaling might enhance anti-tumor chemotherapy (Olive et al, 2009). Thus, disruption of paracrine SHH-SMO signaling through inhibition of stromal SMO emerged as a promising pre-clinical target. However, subsequent clinical trials based on these observations failed in pancreatic cancer patients (Ruch & Kim, 2013). More recently, ablation of SHH ligand in tumor cells was shown to decrease stromal activation and increase tumor cell growth (Lee et al, 2014; Rhim et al, 2014). Consistent with these results, work from our group demonstrated that deletion of the key HH signaling effector Smoothened (Smo) from SMA-positive fibroblasts led to an increase in pre-neoplastic acinar-to-ductal metaplasia (ADM) (Liu et al, 2016). The mechanism involved activation of a non-canonical AKT/GLI2 oncogenic pathway, production of TGF-$\alpha$ by fibroblasts, and activation of epidermal growth factor receptor signaling in epithelial cells (Liu et al, 2016).

In the present work, we provide mechanistic details upstream of AKT activation in Smo-deficient fibroblasts. We demonstrate that loss of SMO results in reduced phosphatase and tensin homolog (PTEN) protein stability that is linked to increased GSK3$\beta$ activity. We identify RNF5 as the E3 ubiquitin ligase targeting PTEN for proteasome-dependent degradation in the absence of SMO. These results indicate that PTEN is a molecular switch that can determine whether stromal fibroblasts act in a suppressive or oncogenic fashion.

## Results

### Disruption of SMO signaling in pancreatic fibroblasts increases PDAC tumor cell growth and decreases stability of PTEN

Whether canonical HH signaling through SMO in pancreatic fibroblasts suppresses or promotes tumor cell growth remains controversial (Olive et al, 2009; Rhim et al, 2014). To directly address this question, we co-injected pancreatic fibroblasts, in which Smo was deleted by Cre/loxP technology, with a luciferase-tagged mouse KPC-luc tumor cell line (derived from LSL-Kras$^{G12D/+}$; TP53$^{loxP/loxP}$; Pdx1-Cre mice [Hwang et al, 2008]) directly into the pancreas of nude mice (see the Materials and Methods section for description of cell lines used). Decreased SMO expression in the pancreatic fibroblasts and expression of Shh in KPC-luc tumor cells was confirmed before injection (Fig S1A and B). After injection, KPC-luc tumor cells were visualized over time via bioluminescence imaging, revealing that KPC-luc tumor cells injected alone or mixed with Smo$^{WT}$ fibroblasts produced tumors of the same size after 15 d (Fig 1A–C). By contrast, KPC-luc cells injected with fibroblasts lacking SMO (Smo$^{KO}$) formed tumors that were significantly larger than controls (Fig 1A and B). Tumor cell growth, as measured by bioluminescence, was significantly increased in KPC-luc cells co-injected with Smo$^{KO}$ fibroblasts relative to Smo$^{WT}$ fibroblasts

(Fig 1C). To confirm these results in a related assay, we co-injected the same fibroblasts with a different mouse tumor cell line, KPC2 (from LSL-Kras$^{G12D/+}$; TP53$^{R172H/+}$; Elas-Cre$^{ER}$ mice), into the flanks of nude mice. Shh expression was confirmed in KPC2 cells before injection (Fig S1B). KPC2 tumor cells injected alone or mixed with Smo$^{WT}$ fibroblasts produced tumors of the same size after 5 wk (Fig S1C and D). Similar to orthotopic injection, flank KPC2 cells co-injected with Smo$^{KO}$ fibroblasts formed tumors that were significantly larger than controls (Fig S1C and D). Further analysis demonstrated an increase in Ki67-positive, proliferating tumor cells upon co-injection with Smo$^{KO}$ fibroblasts relative to Smo$^{WT}$ fibroblasts (Fig S1E and F).

Our previous work demonstrated that activation of AKT upon genetic deletion of Smo in pancreatic fibroblasts accelerated ADM and epithelial cell proliferation (Liu et al, 2016). Whether loss of PTEN expression contributed to the activation of the AKT pathway was studied further. Western blot analysis revealed that PTEN protein was lost and AKT phosphorylation at Ser-473 was increased in Smo$^{KO}$ fibroblasts (Fig 1D). Surprisingly, Pten mRNA levels remained unchanged between Smo$^{WT}$ and Smo$^{KO}$ fibroblasts (Fig 1E).

To address the mechanism by which PTEN protein levels were down-regulated in the absence of Smo, we treated pancreatic fibroblasts with cycloheximide and measured PTEN protein levels over time. PTEN protein was highly stable in control Smo$^{WT}$ fibroblasts and remained unchanged over the 24-h period of cycloheximide treatment (Fig 2A and B, lanes 1–6). Strikingly, PTEN protein levels, even when twice the amount of total protein was loaded on the gel, were dramatically reduced in Smo-deleted fibroblasts by 8–16 h of cycloheximide treatment (Fig 2A and B, lanes 8–13). Cycloheximide treatment led to the expected reduction in TP53 in both Smo$^{WT}$ and Smo$^{KO}$ fibroblasts (Fig 2A). To determine if PTEN degradation was proteasome-dependent, fibroblasts were treated with MG132, a potent proteasome inhibitor. MG132 treatment of Smo$^{KO}$ cells restored PTEN protein to wild-type levels (Fig 2C and D, lanes 5–8), but had no obvious effect on control cells where PTEN protein was already very stable (Fig 2C and D, lanes 1–4).

### PTEN loss in tumor-associated fibroblasts correlates with reduced overall survival in human PDAC patient samples

To test the hypothesis that loss of fibroblast PTEN is driving disease progression, the Vectra multispectral imaging platform was used to analyze PTEN levels in SMA-positive pancreatic fibroblasts in a patient tissue microarray (TMA; representative images in Figs 2E and S2A). In support of using the dual immunohistochemistry (IHC) methodology, the same results were obtained for PTEN/SMA staining with dual IHC compared with dual immunofluorescence (IF) staining (Fig S2B–E). We examined the relationship between patient outcome and reduced PTEN expression in SMA-positive fibroblasts. Patient samples with PTEN expression scores in the lower quartile had significantly poorer overall survival, with median survival of 8.8 mo for the low PTEN group versus 16.9 mo for the group with higher PTEN scores (Fig 2F and Table S1A; log-rank test; P = 0.017). Multivariate Cox regression analysis revealed that low PTEN scores trended toward statistical significance as a predictor of patient survival (Table S1A; P = 0.052). Although multivariate analysis did not reach statistical significance in the entire cohort, univariate

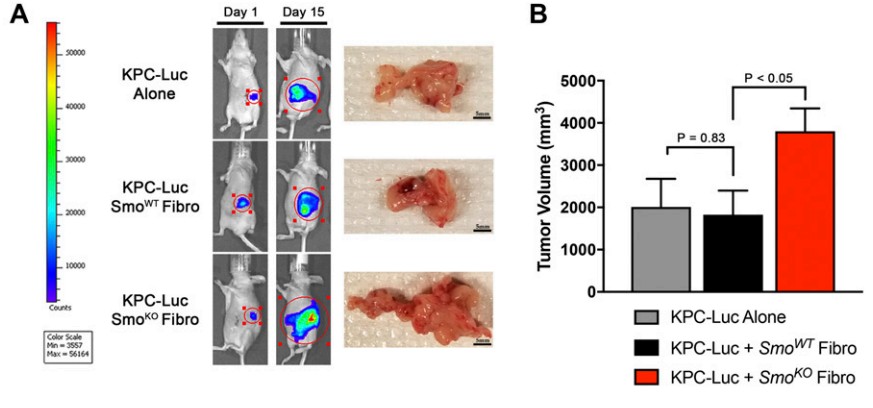

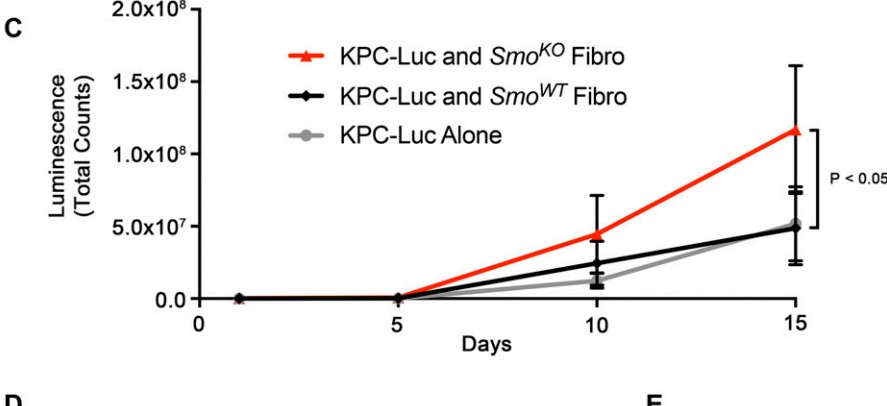

**Figure 1.  *Smo*-null fibroblasts accelerate PDAC tumor cell growth and proliferation in vivo.**
**(A)** Bioluminescence images of KPC-luc tumor cells mixed with $Smo^{WT}$ or $Smo^{KO}$ fibroblasts at day 1 and day 15 post-orthotopic injection. **(B)** Average tumor volume at day 15 post-orthotopic injection. N = 5, bars indicate means ± SD. **(C)** Quantification of bioluminescence in orthotopic injection mice. *P* value calculated using repeated measure ANOVA. **(D)** Western blots and quantification with indicated antibodies in $Smo^{WT}$ versus $Smo^{KO}$ fibroblasts. N = 3, bars indicate means ± SD. **(E)** qRT-PCR analysis of *Pten* in $Smo^{WT}$ versus $Smo^{KO}$ fibroblasts. N = 3, bars represent means ± SD.

subgroup analyses showed that the association between PTEN score and overall survival was statistically significant for the group of patients older than 60 y, but not for those 60 or less (Table S1B and C; *P* = 0.003 and 0.609, respectively). Importantly, reduced PTEN levels remained a significant predictor of overall survival in multivariate analysis of the portion of the patient cohort older than 60 y (Table S1C; *P* = 0.022). Patient cohort demographics for each subgroup are presented in Table S2.

To determine the functional significance of decreased PTEN levels in PDAC patient fibroblasts, the human PDAC cell line MIA PaCa-2 and primary PDAC patient-derived cancer associated fibroblasts (CAFs) with or without *Pten* depletion by shPTEN (PTEN knockdown confirmed in Fig S3A) were admixed and injected into the pancreas of nude mice. MIA PaCa-2 cells injected with shPTEN fibroblasts were larger than those injected with control shNC fibroblasts (Fig 3A and B). Injected human fibroblasts were shown to persist at the time of harvest by staining dual IHC for SMA and a human-specific anti-mitochondria antibody (dual positive SMA+/HumanMito+ cells are indicated by red arrowheads in Fig S3B). Similar to the orthotopic model, xenograft MIA PaCa-2 cells

co-injected with shPTEN fibroblasts also grew at a significantly faster rate than those injected with control shNC fibroblasts (Fig 3C and D). Tumor cells co-injected with shPTEN fibroblasts showed increased proliferation compared with controls, as measured by Ki67 staining of the tumor cells (Fig S3C and D). Furthermore, staining for the endothelial cell marker Meca32 revealed an increase in the number and size of tumor blood vessels (Fig 3E). Similarly, xenograft co-injection of the mouse PDAC cell line KPC2 and mouse $Pten^{WT}$ and $Pten^{KO}$ pancreatic fibroblasts (confirmed knockout in Fig S3E) resulted in increased tumor growth, tumor cell proliferation, and tumor angiogenesis (Fig S3F–I, respectively).

**PTEN stability is modulated by the E3 Ubiquitin Ligase RNF5**

To begin to address the mechanisms by which SMO signaling controls PTEN stability in stromal fibroblasts, an unbiased global PTEN interactome screen was implemented. Purified V5-tagged PTEN was incubated on slides on which 17,000 human GST-tagged proteins were printed as previously described (Jeong

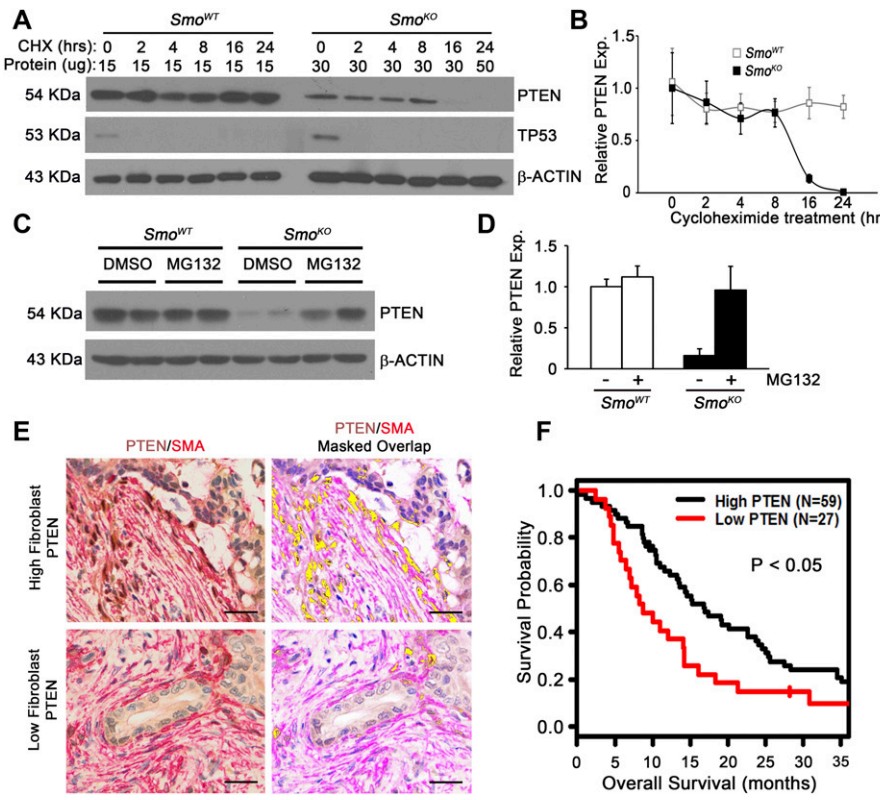

Figure 2. Proteasome-mediated degradation of PTEN in *Smo*-null cells.
**(A)** Western blots for PTEN and TP53 upon cycloheximide treatment in *Smo*<sup>WT</sup> or *Smo*<sup>KO</sup> fibroblasts. Protein loading amount (μg) indicated above lane. **(B)** The graph represents quantification of three independent Western blots relative to untreated. N = 3, squares represent means ± SD. **(C)** Western blots for PTEN in DMSO- (Vehicle) or MG132-treated *Smo*<sup>WT</sup> or *Smo*<sup>KO</sup> fibroblasts. **(D)** Graph represents quantitation of three individual Western blots relative to vehicle-treated. N = 3, bars represent means ± SD. **(E)** Composite images (1 image per core) of dual color IHC (PTEN Brown, SMA Red) of human PDAC TMA and co-localization map showing SMA and PTEN overlap in yellow. Scale bar 50 μm. **(F)** Kaplan–Meier plots for fibroblast PTEN expression (H-score cutoff of 22) Scale bars, 50 μm.

et al, 2012) (see the Materials and Methods section). When a stringent signal-to-noise cutoff ratio of 8.0 was applied, 374 predicted PTEN interacting proteins were identified, including previously identified binding partners (Table S3). As PTEN degradation is mediated by the proteasome in *Smo*-deleted fibroblasts, we focused on potential PTEN E3 ubiquitin ligases, the key enzyme in the pathway that recognizes specific protein substrates for ubiquitination (Lecker et al, 2006). The 374 PTEN interactome list contained only three E3 ubiquitin ligases: TRIM44, RNF5, and WWP2, of which the latter is a known E3 ubiquitin ligase for PTEN (Maddika et al, 2011). Knockdown of the three candidate E3 enzymes (TRIM44, RNF5, and WWP2) in *Smo*<sup>KO</sup> fibroblasts demonstrated that only *Rnf5* silencing significantly rescued PTEN protein levels to wild type (Fig 4A and B). Knockdown efficiency for each gene was confirmed at the RNA level (Fig S4A), and knockdown of RNF5 protein confirmed (Fig S4B). As expected, *Rnf5* knockdown had no effect on *Pten* mRNA levels (Fig S4C). Overexpression of supra-physiological *Pten* in *Smo*<sup>KO</sup> fibroblasts was unable to rescue PTEN protein levels, and phosphorylation of AKT at Ser-473 remained high, indicating that, in this context, RNF5 in these cells is a potent PTEN E3 ubiquitin ligase and rapidly degrades overexpressed PTEN (Fig S4D and E).

Co-immunoprecipitation (co-IP) was performed with PTEN antibody in *Smo*<sup>KO</sup> fibroblasts, followed by Western blot analysis with RNF5 antibody, demonstrating interaction of these two proteins in the context of fibroblast *Smo* deletion (Fig 4C). Remarkably, expression of the combination of V5-tagged RNF5 and HA-tagged ubiquitin in 293T cells decreased endogenous PTEN levels (Fig 4D). In the same experiment, immunoprecipitation of endogenous PTEN

and Western blot analysis with HA-antibody demonstrated RNF5-mediated ubiquitin transfer to PTEN (Fig 4E).

### GSK3β mediates PTEN degradation by phosphorylation of Threonine-366

Previous work demonstrated that PTEN was destabilized by GSK3β phosphorylation at position Threonine-366 in the unstructured C-terminal tail of PTEN (Maccario et al, 2007). Furthermore, GSK3β is a known component of the Hedgehog Signaling Complex that regulates Gli1 stability (Jia et al, 2002; Sharpe et al, 2015). Therefore, we tested whether decreased PTEN stability in *Smo*<sup>KO</sup> cells may be due to phosphorylation of Threonine-366 by GSK3β (Maccario et al, 2007). *Smo*<sup>KO</sup> fibroblasts had higher levels of GSK3β-pTyr-216, the activated form of the kinase (Fig 5A). Furthermore, PTEN-pT366 was detected in *Smo*<sup>KO</sup> cell extracts and the phosphorylated form of the protein accumulated after treatment of cells with the proteasome inhibitor MG-132 (Fig 5B). To directly test whether GSK3β activity was responsible for destabilization of PTEN, *Smo*<sup>KO</sup> fibroblasts were treated with three different GSK3β inhibitors (CT99021, AR-A014418, and SB-216763) over a 48-h time course and PTEN levels were analyzed via Western blot analysis. Inhibition of GSK3β activity was confirmed by showing decreased levels of Glycogen Synthase pSer-614, a well-characterized substrate site for GSK3β (Fig 5C). All three GSK3β inhibitors fully rescued PTEN protein levels by 48 h (Fig 5C and D). In contrast to proteasome inhibition with MG-132 (Fig 5B), PTEN-pT366 did not appreciably accumulate following treatment with the GSK3β inhibitors (Fig 5C), validating its role as a PTEN kinase.

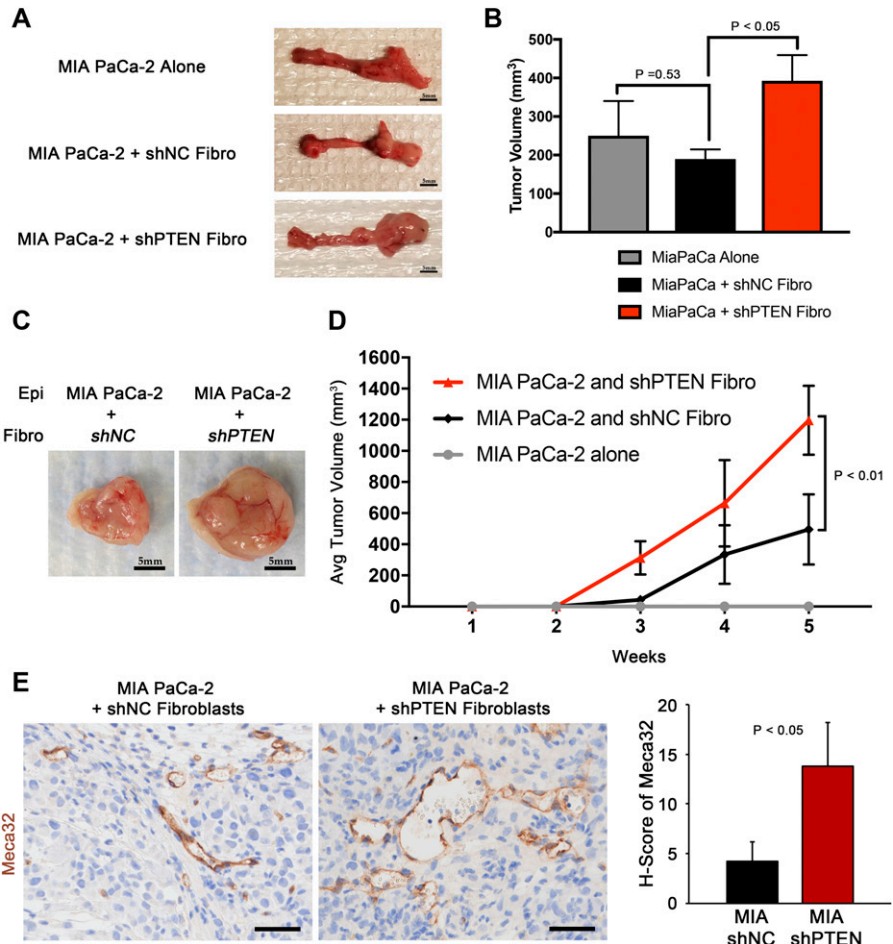

Figure 3. *Pten*-null fibroblasts accelerate PDAC tumor cell growth and proliferation in vivo. **(A, B)** Images and tumor volume quantification of orthotopic MIA PaCa-2 tumor cells co-injected with shNC⁻ (scrambled control) or shPTEN-transduced fibroblasts. N = 5, dots represent means ± SEM. **(C, D)** Images and tumor volume quantification of xenograft MIA PaCa-2 tumor cells co-injected with shNC- (scrambled control) or shPTEN-transduced fibroblasts. N = 5, dots represent means ± SEM. *P*-value calculated using repeated measure ANOVA. **(E)** IHC and quantification of Meca32 staining. N = 3, bars represent means ± SD.

## Pharmacologic inhibition of SMO destabilizes PTEN in fibroblasts leading to decreased hydraulic permeability

SMO antagonists were promising drugs for pancreatic cancer in pre-clinical studies; however, clinical trials with SMO antagonists in combination with chemotherapy failed (Ruch & Kim, 2013). We hypothesized that the down regulation of stromal PTEN levels might contribute to failure of SMO inhibitors such as GDC-0449. To determine if inhibition of SMO with this small molecule inhibitor could mimic the genetic deletion of SMO, *LSL-Kras^{G12D/+};LSL-Trp53^{R270H/+};Pdx-1-Cre;Brca1^{loxP/loxP}* (KPC-BRCA1^{CKO}) mice were treated with GDC-0449 (Shakya et al 2011, 2013). This genetically engineered mouse model (GEMM) was chosen as it is an aggressive stroma-rich GEMM of PDAC. KPC-BRCA1^{CKO} mice were treated with GDC-0449 at 5 wk of age, when invasive ductal adenocarcinoma is detected. Short-term treatment of mice with GDC-0449 was performed as a proof-of-principle experiment to show in an autochthonous GEMM of PDAC that SMO inhibition could lead to decreased PTEN levels in vivo. In addition, long-term treatment with GDC-0449 proved toxic in KPC-BRCA1 mice. Analysis of *Gli1* and *Ptch1* mRNA expression in the tumors validated the efficacy of SMO inhibition (Fig 6A). A significant decrease in the PTEN protein levels in SMA-positive fibroblasts was observed in the GDC-0449 group compared with controls as quantified using multispectral microscopy (Figs 6B and S5A).

PDAC patient-derived primary pancreatic CAF were treated with GDC-0449 and Western blot analysis demonstrated a rapid decrease in PTEN protein and an increase in phosphorylated, activated AKT in response to drug treatment (Figs 6C and S5B). The same experiment was repeated in *Smo^{WT}* mouse pancreatic fibroblasts. GDC-0449 treatment effectively decreased PTEN expression and activated AKT in these mouse fibroblasts (Fig S5C). Of note, human fibroblasts required a slightly higher dose of GDC-0449 to reduce PTEN levels, suggesting that human PTEN may be more stable than mouse PTEN.

Fibroblasts are major contributors to the distinct desmoplastic reaction in PDAC that alter the architecture and mechanics of the ECM (Provenzano et al, 2012; Stylianopoulos et al, 2013). We designed an in vitro assay using a microfluidic device (Hammer et al, 2017) to test whether PTEN in tumor fibroblasts affects the hydraulic permeability (K), which is a characteristic of the ECM that relates interstitial fluid velocity to the fluid pressure gradient (Wiig & Swartz, 2012) (Fig S6A–D). This in vitro assay demonstrated that human pancreatic tumor fibroblasts with *PTEN* knockdown had increased resistance to flow (i.e., decreased K) compared with control (Fig 6D and E). Consistent with our results, GDC-0449 treatment of control fibroblasts, but not PTEN knockdown fibroblasts, showed a similar decrease in K relative to untreated cells

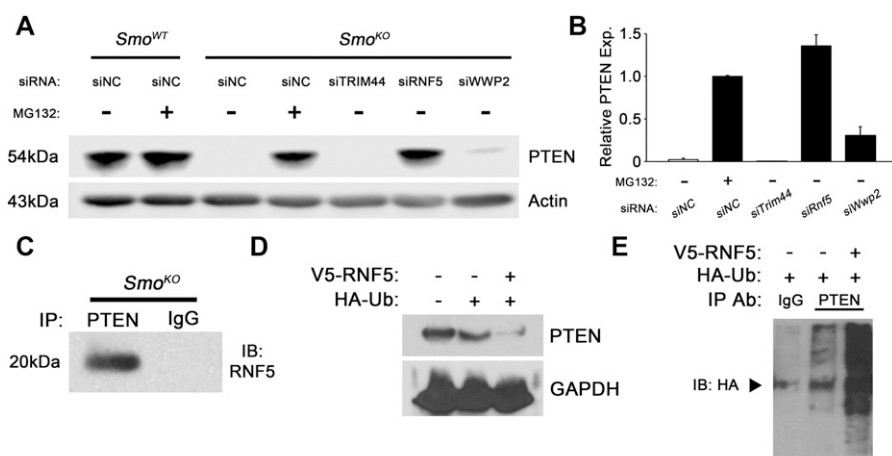

**Figure 4.    RNF5 is a novel E3 ubiquitin ligase for PTEN.**
**(A)** Western blots for PTEN upon treatment with the indicated siRNA or MG132 in *Smo*<sup>WT</sup> or *Smo*<sup>KO</sup> fibroblasts. **(B)** Graph represents quantitation of three individual Western blots relative to vehicle-treated. N = 3, bars represent means ± SD. **(C)** Western blots for RNF5 after co-IP with PTEN or IgG antibody in *Smo*<sup>KO</sup> fibroblasts. N = 3. **(D)** Western blots for PTEN in HEK-293T cells fibroblasts transfected with FLAG-tagged RNF5 or HA-tagged Ubiquitin. N = 3. **(E)** Western blots for HA-Ubiquitin after co-IP with PTEN antibody in HEK-293T cells fibroblasts transfected with FLAG-tagged RNF5 or HA-tagged Ubiquitin. N = 3.

(Fig 6D and E). Increased hyaluronic acid (HA) in the ECM is known to significantly alter the hydraulic permeability in tumors and other tissues (Wiig & Swartz, 2012). Interestingly, treating with hyaluronidase normalized the decreased hydraulic permeability produced by PTEN-knockdown tumor fibroblasts (Fig 6D and E). HA production is regulated by hyaluronan synthase genes *HAS1*, *HAS2*, and *HAS3* (Itano & Kimata, 2002), of which *HAS2* and *HAS3* are expressed by pancreatic fibroblasts. Knockdown of *PTEN* resulted in a significant increase in *HAS3* (Fig 6F), which has previously been shown to promote tumor growth in pancreatic cancer (Kultti et al, 2014). Furthermore, analysis of The Cancer Genome Atlas indicated that increased *HAS3* expression correlated with decreased survival in pancreatic cancer patients (Fig 6G).

## Discussion

We have identified a mechanism by which stromal fibroblasts promote pancreatic tumor cell growth. Previous studies revealed a potential suppressive function of the tumor microenvironment (Lee et al, 2014; Ozdemir et al, 2014; Rhim et al, 2014). In addition, HH signaling has been shown to act in a paracrine manner in PDAC, with tumor-secreted SHH activating the HH pathway in pancreatic fibroblasts (Yauch et al, 2008).These studies, however, did not directly address the potential oncogenic functions of the tumor microenvironment in pancreatic cancer progression. Our study definitively illustrates that a set of fibroblasts within the tumor microenvironment can promote tumor cell growth when paracrine hedgehog signaling is disrupted. Although others have demonstrated that treatment of xenograft tumors with HH antagonists delays tumor growth (Yauch et al, 2008), we crucially extend these findings to demonstrate that *Smo* deletion in fibroblasts enhances tumor growth. These seemingly paradoxical results are likely due to the different origin of fibroblast cultures. Yauch et al (2008) used mouse embryonic fibroblasts with ex vivo cre-mediated recombination, whereas we used CAFs with in vivo genetic deletion of *Smo*. This is in agreement with recent literature demonstrating that activated CAFs are inherently different than resident fibroblasts (reviewed in Kalluri [2016]). Notably, we also provide mechanistic

insight into the events that connect the loss of SMO and PTEN, identifying GSK3β and the E3-ligase RNF5 as critical intermediates in the proteasome-mediated destruction of PTEN (Fig 7). These results extend previous work from our group demonstrating that loss of Hedgehog signaling in pancreatic stromal fibroblasts caused increased PI3-kinase/AKT signaling and non-canonical activation of the GLI2 transcription factor, events that led to enhanced TGFα production by fibroblasts and accelerated ADM and growth of pancreatic tumor cells via activation of epidermal growth factor receptor signaling (Fig 7) (Liu et al, 2016).

Analysis of fibroblast PTEN expression in PDAC patient samples demonstrated heterogeneous patterns of expression in the stromal fibroblast compartment, with focal areas of intense staining intermixed with areas of minimal staining in the same patient tissue sample. Given this observation, we were intrigued to find that patients who had decreased stromal PTEN correlated with a worse prognosis (Fig 2E and F). Importantly, we selected multiple areas of tissue for this survival analysis, to obtain a representative and quantifiable measure of the global stromal PTEN expression for each patient, to control for heterogeneity in staining patterns. Moreover, recent results demonstrating distinct sets of activated fibroblasts that either produce or respond to IL6 highlight the importance of stromal heterogeneity in promoting pancreatic tumor progression (Ohlund et al, 2017). However, taken together, we caution the clinical interpretation of results presented in this manuscript, as analyses of TMA staining is limited with regard to intratumoral heterogeneity. Heterogeneity of tumor cells has long been appreciated and contributes to tumor recurrence and therapeutic resistance. Molecular heterogeneity within the stromal fibroblast population may account for, at least in part, the tumor-promoting and tumor-suppressive fibroblast subsets within pancreatic cancer. The combined results begin to establish the complex heterogeneity of pancreatic CAFs and the potential functional consequences of modulating the stroma with targeted therapies. The work presented herein establishes that loss of stromal PTEN influences tumor growth in a non-cell autonomous fashion.

Structural alterations to the ECM mediated by PTEN expression in stromal fibroblasts was assessed by quantifying K, which is a widely used measurement in interstitial physiology (Wiig & Swartz, 2012).

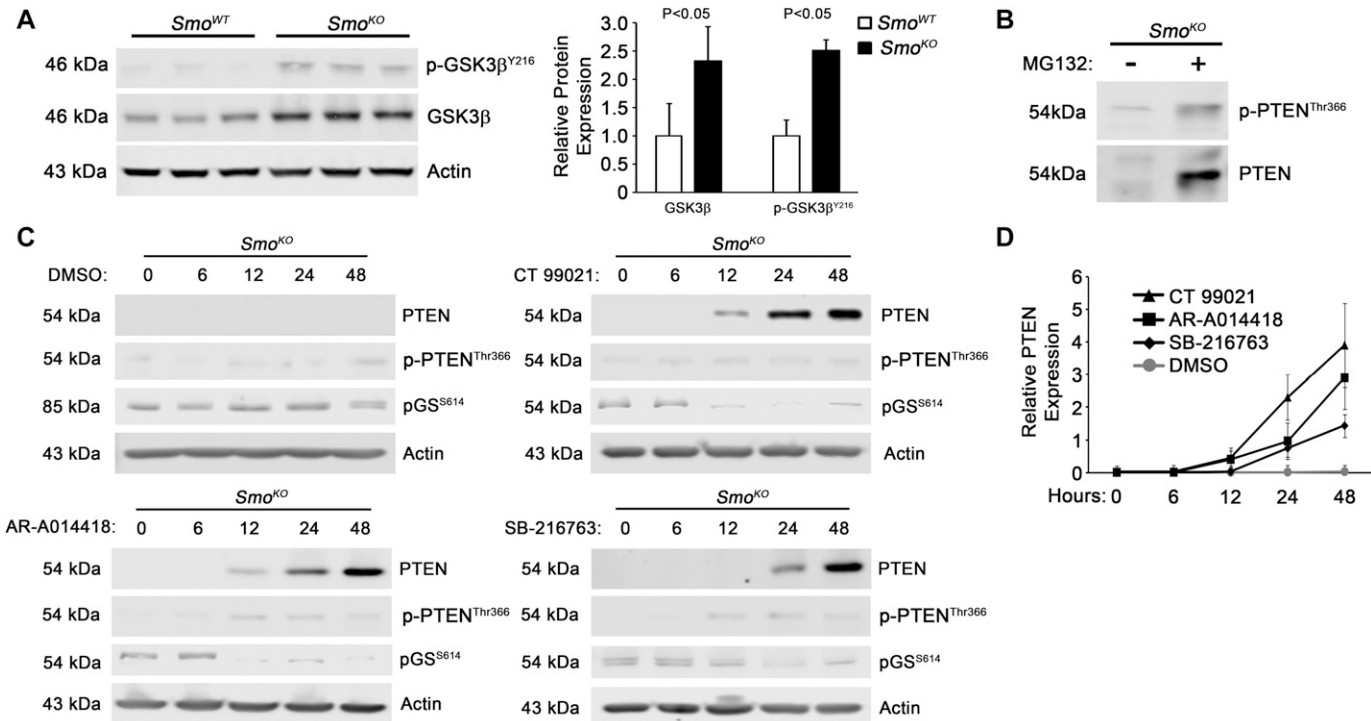

**Figure 5. GSK3β inhibition blocks PTEN degradation.**
**(A)** Western blots and quantification for total and phosphorylated GSK3β Tyr-216 $Smo^{KO}$ or $Smo^{KO}$ fibroblasts. Graph represents quantitation of three individual Western blots relative to $Smo^{WT}$. N = 3, bars represent means ± SD. **(B)** Western blots of total and phosphorylated PTEN Thr-366 upon treatment with vehicle or MG132 in $Smo^{KO}$ fibroblasts. N = 3. **(C, D)** Western blots and quantification for total and phosphorylated PTEN Thr-366 and phosphorylated glycogen synthase (GS) Ser-614 in $Smo^{KO}$ or $Smo^{KO}$ fibroblasts at the indicated time points. Graph represents quantitation of three individual Western blots relative to $Smo^{WT}$. N = 3, dots represent means ± SD.

Our results suggest that loss of PTEN in fibroblasts correlates with increased HA synthesis that results in decreased hydraulic permeability and is a subsequent physical barrier to interstitial drug transport in tumors. Hingorani et al, have demonstrated that targeting HA with PEGylated, recombinant human hyaluronidase increases the effectiveness of chemotherapy in pre-clinical PDAC models and in PDAC patients (Provenzano et al, 2012; Hingorani et al, 2016). Furthermore, the original study by Olive et al, demonstrated that treatment with SMO inhibitors led to increased intratumoral vasculature (Olive et al, 2009), in agreement with our result that PTEN silencing in fibroblasts enhanced angiogenesis. This result is intriguing and suggests that fibroblasts are integral components of the microenvironment that regulate angiogenesis and the vascular network, and that clinical interventions targeting the stroma should consider potential effects this may have on tumor vasculature.

Importantly, the identification of RNF5 as a new E3 ubiquitin ligase for PTEN shows the cell-type specificity of proteasome-mediated degradation machinery. Several E3 ubiquitin ligases that mediate PTEN degradation have been previously identified, including NEDD4-1, WWP2, and XIAP (Wang et al, 2007; Van Themsche et al, 2009; Maddika et al, 2011). However, of these three, only WWP2 was identified in our initial PTEN interactome screen and subsequent experiments indicated that WWP2 is not responsible for degradation of PTEN in pancreatic stromal fibroblasts. Given this result, future studies will be required to determine if RNF5 acts as

an E3 ubiquitin ligase in other cell types and cancers. Breast cancer, in particular, has been previously shown to over-express RNF5 (Bromberg et al, 2007), whereas other cancers have yet to be explored. Of note, 75% of prostate cancer patients with reduced PTEN protein lack a corresponding reduction in mRNA, emphasizing that PTEN protein decay mechanisms may have a broader context within human cancers (Chen et al, 2011). Therefore, targeting PTEN destabilizers such as RNF5 may provide a unique approach to restore PTEN function in both tumor cells and tumor stromal fibroblasts.

# Materials and Methods

### Animal strains and maintenance

Control ($Smo^{loxP/-}$, herein referred to as $Smo^{WT}$) and experimental ($FspCre;Smo^{loxP/-}$, herein referred to as $Smo^{KO}$) animals were generated by crossing the previously described $Smo^{loxP}$, $Smo^-$ (Long et al, 2001) and $FspCre$ (Trimboli et al, 2008, 2009) strains with tumor-bearing $Mist1-Kras^{G12D}$ animals, as previously described by our group (Liu et al, 2016; Pitarresi et al, 2016). Pancreatic fibroblasts were generated from age-matched littermate $Smo^{WT}$ and $Smo^{KO}$ mice from a mixed C57BL/6; 129/Sv and FVBN genetic background; all cells were passaged the same number of times for each experiment. $LSL-Kras^{G12D/+};LSL-Trp53^{R270H/+};Pdx-1-Cre;Brca1^{loxP/loxP}$

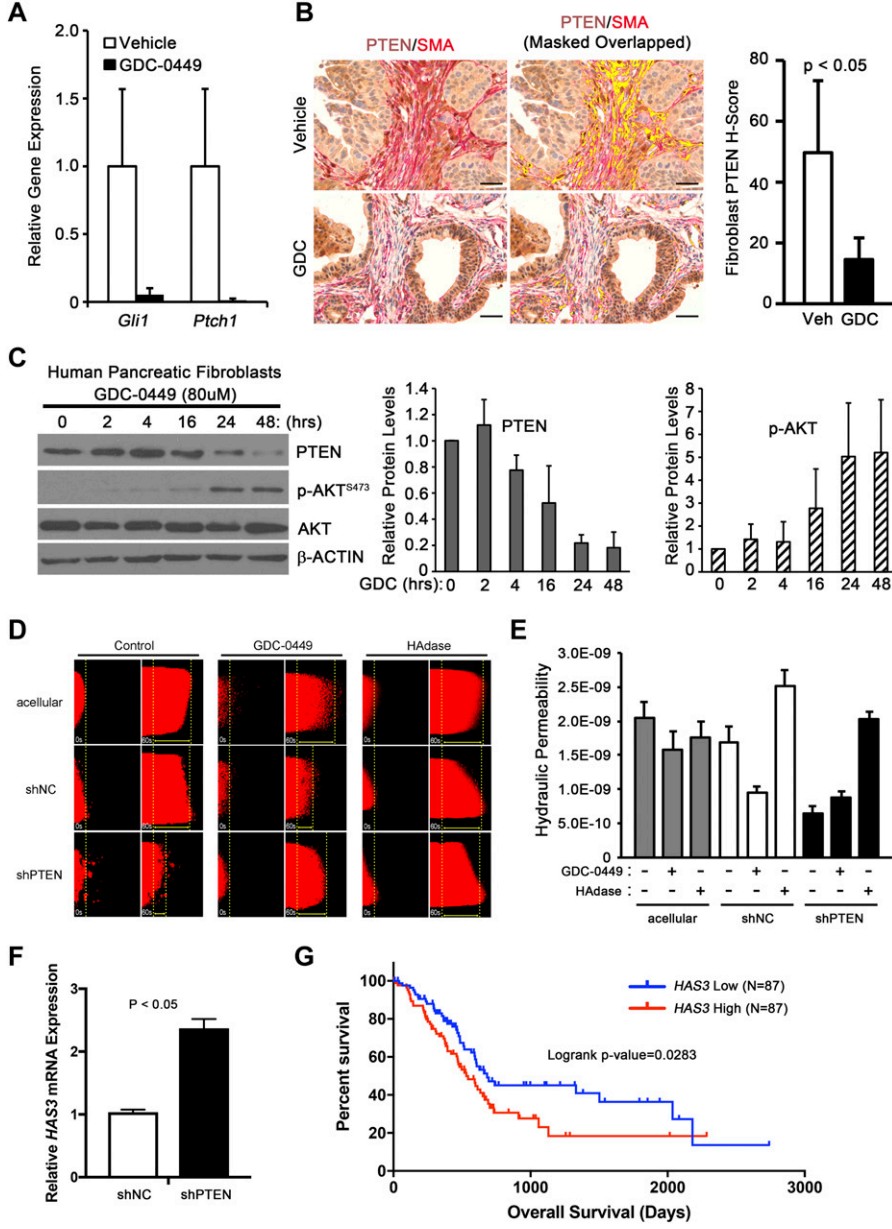

**Figure 6. Pharmacological Inhibition of SMO destabilizes PTEN.**
**(A)** qRT-PCR analysis of whole pancreas tissue for *Gli1* and *Ptch1* in vehicle or GDC-0449 treated mice. N = 3, bars represent means ± SD. **(B)** Dual color IHC (PTEN brown, SMA red) of pancreata from vehicle or GDC-0449 treated mice, showing SMA and PTEN overlap in yellow. Scale bar, 50 *μm*. The H-score represents the quantification of the PTEN staining in SMA-positive cells. N = 3, bars represent means ± SD. **(C)** Western blots and quantification for total PTEN, total and phosphorylated AKT Ser-473 in GDC-0449–treated human pancreatic CAFs. Graph represents quantitation of three individual Western blots relative to untreated. N = 3, bars represent means ± SD. **(D)** Rhodamine-BSA dye flow through collagen channel after 48-h culturing of indicated fibroblast populations with and without GDC-0449. Yellow dotted line indicates displacement of dye at 60 s relative to 0 s. **(E)** Quantification of hydraulic permeability (rate shown in E). Statistical testing was performed using ANOVA with Tukey's HSD post-testing. N ≥ 4 for each condition. Bars indicate means ± SEM; ANOVA with Tukey's–Cramer post-testing was used to analyze statistical significance. **(F)** qRT-PCR analysis of *HAS3* in shNT and shPTEN fibroblasts. N = 3, bars represent means ± SD. **(G)** Kaplan–Meier survival curve of The Cancer Genome Atlas-PAAD data segregated based on *HAS3* high or low expression.

(KPC-BRCA1[CKO]) mice were previously described (Shakya et al, 2013) and generously provided by Reena Shakya and Thomas Ludwig.

GDC-0449 was dissolved in DMSO and mice were orally administered 100 mg/kg once per day for 4 d. 4 h after the last dose, mice were dissected and pancreata were harvested for histology and immunohistochemistry staining as described.

### Xenograft and orthotopic injections

Athymic nude mice used for xenograft injection experiments were provided by the Ohio State University Target Validation Shared Resource Core. For subcutaneous injection, $5 \times 10^5$ KPC2 or MIA PaCa-2 tumor cells were admixed in a 1:1 ratio with $5 \times 10^5$ fibroblasts and injected into the flanks of nude mice; tumor volumes were measured once per week by caliper.

Athymic nude mice were used for orthotopic injection experiments. For mouse cell lines, $1 \times 10^5$ KPC-luc tumor cells were admixed in a 1:5 ratio with $5 \times 10^5$ fibroblasts, as previously described (Hwang et al, 2008). KPC-luc cells were generously provided by Craig D. Logsdon and Zobeida Cruz-Monserrate. For human cell lines, $1 \times 10^6$ MIA PaCa-2 tumor cells were admixed in a 1:5 ratio with $5 \times 10^6$ fibroblasts. KPC-luc tumor-bearing mice were injected with 15 mg/ml of D-luciferin (IP) and imaged every 5 d for the course of the study on a PerkinElmer IVIS Spectrum imaging system with the help of the Small Animal Imaging Facility at the University of Pennsylvania. Fibroblasts were shown to persist 6-wk post-injection by staining with SMA and human nuclear antigen in consecutive sections from MIA PaCa-2 tumor cells co-injected with human patient–derived fibroblasts (Fig S3E), in agreement with results from others showing that fibroblasts persist in both

▶▶▶▶ Life Science Alliance

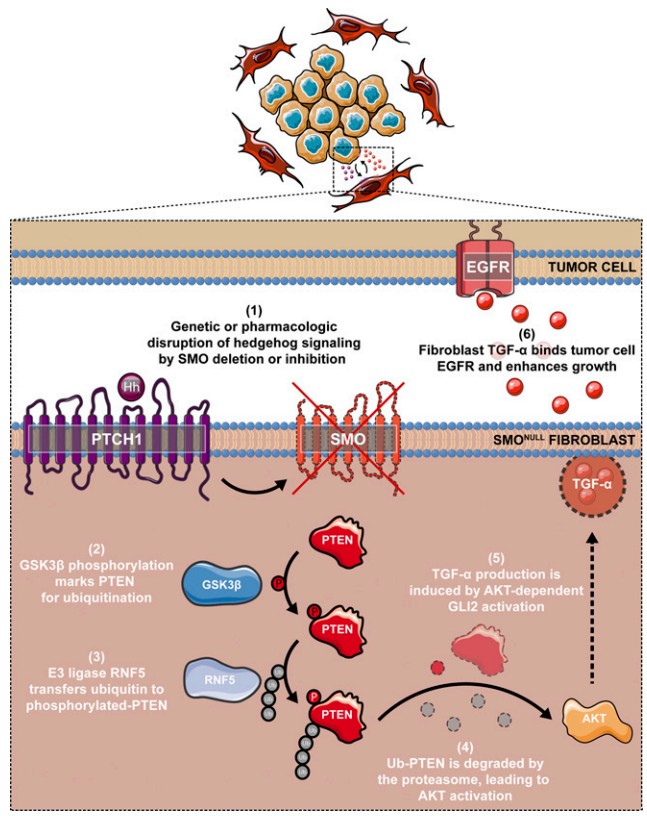

**Figure 7. Graphical abstract.**
Genetic deletion of *Smo* or pharmacologic inhibition of SMO in pancreatic CAFs activates GSK3β, leading to enhanced PTEN phosphorylation and proteasomal degradation via the E3 ubiquitin ligase enzyme RNF5. Subsequent AKT activation leads to enhanced GLI2 binding and activation of the *Tgfa* promoter. TGF-α production by SMO-null fibroblasts cross-talks with the adjacent tumor cells, leading to activation of epidermal growth factor receptor (EGFR) and accelerated growth of the tumor epithelium.

xenograft (Mathew et al, 2014) and orthotopic (Vonlaufen et al, 2008) injection model systems. To confirm that fibroblasts were not transformed, $5.0 \times 10^5$ fibroblasts were injected into the flanks of nude mice and followed for 6 wk; no tumors formed.

## Multispectral vectra IHC analysis

Dual stained samples were imaged using the PerkinElmer's Vectra multispectral slide analysis system. For the mouse samples, at least three multispectral images per animal for at least three mice per genotype (unless otherwise noted) were manually taken. For the human PDAC TMA, one field of interest per core was automatically acquired. The image acquisition workflow consisted of the following: (1) monochrome imaging of the entire slide; (2) RGB low power imaging of the tumor tissue using an inForm tissue finding algorithm; and (3) multispectral high-power imaging of one field containing tumor epithelium and stroma by means of an inForm HPF finding algorithm.

For visualization of the component images (PTEN/SMO DAB, SMA Red), the multispectral images were spectrally unmixed using Nuance software. Nuance co-localization tool was used to create the co-localization maps displaying the PTEN/SMO–positive cells in the SMA-positive cellular compartment overlap in yellow.

For quantification of the PTEN and SMO staining, the multispectral images were reviewed and analyzed using inForm Tissue Finder software. A pattern recognition algorithm was used for processing as follows: (1) trainable tissue segmentation to segment the SMA-positive regions from the tumor epithelium; (2) cell segmentation of the SMA-positive tissue category to locate the subcellular compartments; and (3) scoring to bin the spectrally unmixed DAB signal into four categories depending on the staining intensity (0+, 1+, 2+ and 3+), providing data in percent. The H-Score, which ranges from 0 to 300, was calculated using the following formula: [1 × (% cells 1+) + 2 × (% cells 2+) + 3 × (% cells 3+)]. Thus, H-score measures staining intensity and percentage of positive cells in a given cellular compartment. Comparison of co-IF with dual-color IHC was performed on serial sections of the same tissue to show that IHC is able to recapitulate conventional staining procedures in a quantitative manner (Fig S2A–B). The percentage of SMA/PTEN dual-positive cells by co-IF was quantified (Fig S2C) and compared with H-score (Fig S2D) for the same mice presented in Fig 3C. Of note, the overlap in IHC staining is a direct quantitative measure of staining for one antigen (in the case of this example, PTEN) within a distinct cell population defined by a second antigen (in this case, SMA), and is not simply the overlap in signal between the two stains.

## IHC and IF staining

Dissected mouse pancreas tissues were fixed in 10% neutral-buffered formalin solution for 48 h and transferred to 70% ethanol. Tissues were processed, embedded in paraffin, cut into 5-µm sections on positively charged slides, de-paraffinized, rehydrated, and stained with H&E.

For immunohistochemistry, all sections were stained using a Bond Rx autostainer (Leica) or Roche Discovery ULTRA autostainer, unless otherwise noted, according to the manufacturers recommendations. Antibodies for the following markers were diluted in Antibody diluent (Leica): αSMA (1:4,000; Abcam), Ki67 (1:200; Abcam), PTEN (1:150 Cell Signaling), SMO (1:200; Bioss), Meca-32 (1:50; BD Pharmingen), and human anti-mitochondria (1:500; Abcam).

## Cell culture and treatments

Primary pancreatic fibroblasts were purified as previously outlined in the literature (Liu et al, 2016; Pitarresi et al, 2016). *Smo^WT* and *Smo^KO* pancreatic fibroblast cultures were established from *Mist1^KrasG12D* mice at the PanIN stage that carried *Smo^LoxP* and *FspCre;Smo^LoxP* alleles, respectively. All fibroblast cultures were shown to be SMA-positive in our previous work {Liu et al, 2016 #11; Pitarresi et al, 2016 #32}. Primary human PDAC CAFs were isolated as previously described {Liu et al, 2016 #11}. For isolation of murine and human primary fibroblasts, tumor tissue was minced and digested with collagenase while shaking for 1 h at 37°C. Digested tissue was gravity purified for 10 min in media, and subsequent pellets were subjected to two more gravity purifications, and then seeded on tissue culture dishes. MIA PaCa-2 tumor cell line was obtained from the ATCC. The KPC2 tumor cell line was a generous gift from Stephen F. Konieczny and established from tumor-bearing *Elas-Cre^ER; LSL-Kras^G12D/+; TP53^R172H/+* mice.

For cycloheximide assays, a final concentration of 10 μg/ml was used. Cellular lysates were collected with RIPA buffer at the indicated time points for standard Western blotting analysis.

Proteasome inhibitor MG132 treatment was performed at a final concentration of 10 μM for the indicated time points.

Three GSK3β inhibitors were used: SB-216763 (#sc-200646; Santa Cruz), CT99021 (#CHIR99021; Sigma-Aldrich), and AR-A014418 (#ALX-270-468-M001; Enzo Lifesciences). Cells were grown to 70% confluency and treated with 5 μM SB-216763, 5 μM CT99021, or 10 μM AR-A014418 for the indicated time points. After treatment, cells were harvested for Western blot analysis in RIPA buffer.

### co-IP and ubiquitin assays

Cells were harvested by trypsinization and lysed in non-denaturing extraction buffer (CST 9803), and immunoprecipitation with antibody against PTEN (1:100 CST). After incubation, PRO-A magnetic beads (LSKMAGA02; Millipore) were added. Samples were washed and Laemmli buffer added for Western blot analysis.

### siRNA knockdown

Dharmacon ON-TARGETplus SMARTpool siRNA system was used for knockdown. Briefly, ~60% confluent pancreatic fibroblast cultures were treated with 200 pM of pooled siRNA and 10 μl of lipofectamine 2000 in 600 μl of serum-free OPTI-MEM media for 8 h. The medium was changed to 10% FBS-DMEM and allowed to sit for 24–48 h, at which time the cells were harvested.

### Microfluidic device fabrication

Hydraulic permeability measurements were acquired using microfluidic devices fabricated out of polydimethylsiloxane using soft lithography techniques. The polydimethylsiloxane devices consisted of a single straight channel (5 mm long, 500 μm wide, and 1 mm tall) with 4 mm inlet/outlet ports and irreversibly sealed to a glass slide using plasma oxidation (Harrick, 90 s). For application of flow, a pipet tip (Redi-Tip) was inserted at one of the ports. Tips were cut to ~2–2.4 cm to ensure a tight fit at the port. The microdevices were sterilized with 30 min of UV treatment.

### Fibroblast cell preparation for hydraulic permeability measurements

Primary human pancreatic tumor fibroblasts were maintained in high glucose DMEM supplemented with 10% fetal bovine serum, 1% penn-steptomycin, 0.2% plasmocin, and 0.1% fungin. Rat tail type I collagen gel (Corning Life Science, 6 mg/ml, pH = 7.4), containing fibroblasts at $1.8 \times 10^6$ cells/ml, was introduced into the microfluidic device at 4°C and polymerized at 37°C for 24 h. The collagen concentration was selected to minimize collagen contraction by fibroblasts. In addition, the devices were incubated with fibronectin (100 μg/ml, 30 min) before injection to improve hydrogel adhesion to the channel walls. For GDC treatment conditions, the fibroblasts were resuspended in medium containing GDC at 80 μM GDC before mixing with collagen gel solution and cultured for 48 h with additional GDC containing medium after injection into the microdevices.

The following experimental conditions were conducted in the microfluidic devices: (1) acellular collagen; (2) collagen containing control tumor fibroblasts (shNC); (3) collagen containing tumor fibroblasts with PTEN knocked down by shRNA (shPTEN); (4) collagen containing shnc fibroblasts treated with GDC (shNC + GDC); (5) collagen containing shPTEN fibroblasts treated with GDC (shPTEN + GDC). All devices were cultured for 48 h before measurements by placing approximately 400 μl of medium at the ports and device surface, with the medium being renewed every 24 h.

### Microfluidic hydraulic permeability measurements

To measure hydraulic permeability, a rhodamine-bovine serum albumin (rhodamine-BSA) (Molecular Probes) was flowed through the microfluidic device. Flow was established by applying a fluidic height difference (2–2.4 cm) between the ports using cell culture medium, measured for each sample. After establishing flow, 4–8μL of rhodamine-BSA was injected into the tip and was transported by the flow through the microdevice. Timelapse microscopy experiments were recorded with an epifluorescence Nikon TS-100F microscope equipped with a Q-Imaging QIClick camera. Images were acquired every 15 s for 20–30 min. These images were then used to quantify the average velocity of the dye through the collagen/fibroblast matrix by tracking the position of the bulk of the dye as it flowed through the microdevice using FIJI. Hydraulic permeability was then calculated by using Darcy's Law for flow through porous medium as follows:

$$K = \frac{\mu v \Delta L}{\Delta P}, \tag{1}$$

where μ is the viscosity of the cell culture medium (approximated using water), v is the average fluid velocity, ΔL is the length of the channel, and ΔP is the pressure difference across the channel due to the fluidic height difference and is given by the following equation:

$$\Delta P = \rho g h, \tag{2}$$

where ρ is the density of the cell culture medium (approximated using water), g is the acceleration due to gravity (9.81 m/s$^2$), and h is the fluidic height difference between ports.

### Statistics

Pearson's correlation, Wilcoxon rank-sum test, ANOVA and t test were calculated using R 3.0.1 or Prism. The P values from t tests are listed unless otherwise specified. In all graphs, median, means (bar), and standard deviations or standard error of the means (lines) are denoted. Microarray data were processed by Robust Multi-array Average (RMA) method and analyzed using the moderated t test approach (Yu et al, 2011). For survival analysis, the Kaplan–Meier method and log-rank test were applied to univariate analysis and Cox regression models were used for multivariate analysis. Comparison wise P-value of 0.05 was considered significant.

### Study approval

The use of animals was in compliance with federal and Ohio State University Laboratory Animal Resources regulations.

# Supplementary Information

# Acknowledgments

We acknowledge the Solid Tumor Biology Histology Core Microscopy, Transgenic/Knockout, Target Validation, Analytic Cytometry, Bioinformatics, and Biostatistics Shared Resources. This study was supported by National Institutes of Health grants PO1 CA097189 (MC Ostrowski and G Leone), NRSA F31 CA189757 (JR Pitarresi), NRSA F32 CA221094 (JR Pitarresi), American Cancer Society IRG-67-003-50 (JW Song), American Heart Association 15SDG25480000 (JW Song), and R01 CA124586 (SF Konieczny). This work was also supported by the Department of Defense (W81XWH-14-1-0040, GM Sizemore), Pelotonia Fellowship Program (JR Pitarresi, GM Sizemore, A Avendano, JJ Chang, and CS Ennis), and the OSU Institute for Materials Research. Any opinions, findings, and conclusions expressed in this material are those of the author(s) and do not necessarily reflect those of the Pelotonia Fellowship Program.

### Author Contributions

JR Pitarresi: conceptualization, data curation, formal analysis, funding acquisition, investigation, visualization, methodology, and writing—original draft, review, and editing.
X Liu: data curation, formal analysis, and investigation.
A Avendano: data curation, formal analysis, investigation.
KA Thies: data curation and formal analysis.
GM Sizemore: data curation and formal analysis.
AM Hammer: data curation and formal analysis.
BE Hildreth III: data curation and formal analysis.
DJ Wang: methodology.
SA Steck: data curation and formal analysis.
S Donohue: data curation and formal analysis.
MC Cuitiño: data curation and formal analysis.
RD Kladney: data curation and formal analysis.
TA Mace: data curation and formal analysis.
JJ Chang: data curation and formal analysis.
CS Ennis: data curation.
H Li: data curation.
RH Reeves: data curation and methodology.
S Blackshaw: data curation and methodology.
J Zhang: formal analysis.
L Yu: formal analysis.
SA Fernandez: formal analysis.
WL Frankel: formal analysis.
M Bloomston: data curation.
TJ Rosol: formal analysis.
GB Lesinski: formal analysis.
SF Konieczny: formal analysis.
DC Guttridge: formal analysis.
AK Rustgi: formal analysis.
G Leone: formal analysis and project administration.
JW Song: formal analysis and project administration.
J Wu: formal analysis and project administration.
MC Ostrowski: conceptualization, formal analysis, funding acquisition, investigation, methodology, project administration, and writing—original draft, review, and editing.

### Conflict of Interest Statement

The authors declare that they have no conflict of interest.

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
