## [Reviewer comments · Life Science Alliance]

Disruption of stromal hedgehog signaling initiates RNF5-mediated proteasomal degradation of PTEN and accelerates pancreatic tumor growth

Jason R. Pitarresi, Xin Liu, Alex Avendano, Katie A. Thies, Gina M. Sizemore, Anisha M. Hammer, Blake E. Hildreth, David J. Wang, Sarah A. Steck, Sydney Donohue, Maria C. Cuitiño, Raleigh D. Kladney, Thomas A. Mace, Jonathan J. Chang, Christina S. Ennis, Huiqing Li, Roger H. Reeves, Seth Blackshaw, Jianying Zhang, Lianbo Yu, Soledad A. Fernandez, Wendy L. Frankel, Mark Bloomston, Thomas J. Rosol, Gregory B. Lesinski, Stephen F. Konieczny, Denis C. Guttridge, Anil K. Rustgi, Gustavo Leone, Jonathan W. Song, Jinghai Wu, Michael C. Ostrowski
DOI: 10.26508/lsa.201800190

Review timeline:

Submission Date:	10 September 2018
Editorial Decision:	11 September 2018
Revision Received:	11 October 2018
2 nd Editorial Decision:	15 October 2018
2 nd Revision Received	16 October 2018
Accepted:	17 October 2018

Report:

(Note: Letters and reports are not edited. The original formatting of letters and referee reports may not be reflected in this compilation.)

Please note that the manuscript was previously reviewed at another journal and the reports were taken into account in the decision-making process at Life Science Alliance. Since the original reviews are not subject to Life Science Alliance's transparent review process policy, the reports and author response cannot be published.

1st Editorial Decision

11 September 2018

Thank you for transferring your manuscript entitled "Disruption of stromal hedgehog signaling initiates RNF5-mediated proteasomal degradation of PTEN and accelerates pancreatic tumor growth." to Life Science Alliance. The manuscript was assessed by expert reviewers at another journal before, and the editors have transferred those reports to us with your permission.

The reviewers found the topic of your work important, but would have expected further reaching insight. The reviewers also provided constructive input on how to further strengthen your data. Based on the reports already at hand, we would like to invite you to submit a revised version of your manuscript for publication in Life Science Alliance. Further reaching insight is not needed, but as also discussed with you prior submission to our journal we would expect:

- a point-by-point response to the concerns raised and accordingly text changes / further discussion
- a re-analysis of the human tissue data sets (reviewer #1); and addressing the minor points of this reviewer by expanding the introduction slightly
- that you make sure that sufficient information on the experimental procedures are given (reviewer #2)
- improving stainings for Fig.S3B (reviewer #2)
- toning-down the conclusions on the link to hyaluronic acid and hydraulic permeability (reviewer #2);
- inclusion of a positive control to show the ectopic expression of PTEN (reviewer #2)
- another concern of both reviewers pertained to the GEMM data. You already outlined that due to toxicity problems, longer treatments are not possible. Please mention and discuss this in your manuscript.

Thank you for this interesting contribution to Life Science Alliance. We are looking forward to receiving your revised manuscript.

2nd Editorial Decision

15 October 2018

Thank you for submitting your revised manuscript entitled "Hedgehog signaling disruption leads to RNF5-mediated degradation of PTEN and accelerates PDAC growth". I appreciate the introduced changes and would be happy to publish your paper in Life Science Alliance pending final revisions necessary to meet our formatting guidelines.
